# The Impact of COVID-19 on the Working Equid Community: Responses from 1530 Individuals Accessing NGO Support in 14 Low- and Middle-Income Countries

**DOI:** 10.3390/ani11051363

**Published:** 2021-05-11

**Authors:** Isabella Wild, Amy Gedge, Jessica Burridge, John Burford

**Affiliations:** 1World Horse Welfare, Anne Colvin House, Snetterton, Norwich NR16 2LR, UK; amygedge@worldhorsewelfare.org; 2School of Veterinary Medicine and Science, University of Nottingham, Sutton Bonington Campus, Sutton Bonington, Loughborough LE12 5RD, UK; svyjab@nottingham.ac.uk (J.B.); John.Burford@nottingham.ac.uk (J.B.)

**Keywords:** coronavirus, equine welfare, human–animal relationships, working equids, working animals, LMIC, sustainable development, One Welfare

## Abstract

**Simple Summary:**

The COVID-19 pandemic was declared on 11 March 2020. Countries have been impacted variably, with differing disease control measures implemented. The working equid community includes some of the world’s most marginalised people, who rely on animals for their daily lives and livelihoods. The aim of this study was to learn about the effects of the pandemic on the working equid community, and in doing so, to develop credible methods to collect data in future unprecedented events. There were 1530 survey respondents from a population of individuals who received support from equid welfare projects across 14 low- or middle-income countries projects during November and December 2020. The main findings were that, compared to prior to the pandemic, equids were working less, individuals were receiving less income, with expenses staying the same or increasing. In the short term, different indicators show that the effect on equine welfare has been inconsistent, but most owners reported no change in their equid’s health. However, it is predicted that there will be negative long-term impacts on human and equid welfare due to financial insecurity. This requires monitoring. Collaboration with humanitarian organisations, governments, and animal welfare non-governmental organisations is required to mitigate deep-rooted issues involving the working equid community.

**Abstract:**

The COVID-19 pandemic was declared on 11 March 2020. The working equid community includes some of the world’s most marginalised people, who rely on animals for their daily lives and livelihoods. A cross-sectional study investigated the effects of COVID-19 on working equid communities, with the intention of developing methods for replication in future unprecedented events. A multi-language survey was developed, involving 38 predominantly closed questions, and carried out face-to-face, over telephone, or online. There were 1530 respondents from a population of individuals who received support from equid welfare projects across 14 low- or middle-income countries projects during November and December 2020. Overall, at the time of survey completion, 57% (875/1522) of respondents reported that their equids were working less, 76% (1130/1478) reported a decreased monthly income from equids, and 78% (1186/1519) reported a reduction in household income compared to pre-pandemic levels. Costs of equid upkeep remained the same for 58% (886/1519) of respondents and 68% (1034/1518) reported no change in the health of their equid. The potential long-term impacts on human and equid welfare due to reported financial insecurities necessitates monitoring. A One Welfare approach, involving collaboration with governments, humanitarian, and animal welfare non-governmental organisations is required to mitigate deep-rooted issues.

## 1. Introduction

The rapid spread of SARS-CoV-2 (COVID-19), declared as a pandemic on 11 March 2020 [1], has caused serious health issues and/or death in millions of people [2]. There have been knock-on socioeconomic consequences globally [3]. The World Bank estimates that the pandemic will push 119–124 million people into extreme poverty [4].

It is considered that there are approximately 100–112 million working equids, allied to the lives of 600 million people [5,6,7], this is likely an underestimate [8]. Working equid owners are often poor [9], with a recent study showing two-thirds of owners in Colombia living below the International Poverty Line with an income less than $1.90 a day [10]. Equids are involved directly and indirectly in income-generating activities, with varied roles among communities, countries, and regions [11]. Multiple studies have shown the livelihood benefits of owning a working equid [12,13,14,15,16,17]. Focus groups involving Ethiopian, Kenyan, Indian, and Pakistani women found that 77% ranked equids as their most valuable livestock, with the animals involved in generating income, helping with household chores, collecting food and water, and elevating social status [18]. Working equids improve food security in some of the poorest places [19]. For example, in a region in India, equid-associated income generation enabled the purchase of almost 80% of annual human food requirements [11]. More broadly, working equids have positive impacts on the United Nations Sustainable Development Goals (SDGs) [20].

Welfare problems in working equids are common. This is due to the tough environmental conditions in which they work, the physically demanding workload, and the lack of capacity, income, and education of owners to meet good welfare standards. Animal welfare should be considered alongside strengthening livelihoods and human welfare, taking a One Welfare approach [21,22]. This holistic approach is similar to that of One Health, with its importance highlighted by the zoonotic disease, COVID-19 [23,24]. It has been shown improving animal welfare is inextricably intertwined with achieving the United Nations SDGs [25].

World Horse Welfare, an international Non-Governmental Organisation (NGO), works in partnership with in-country projects to find local, sustainable solutions to improve equid welfare [26]. The worldwide spread of COVID-19 has disrupted project interventions because the need for physical distancing and isolation has reduced contact of project staff, veterinary services, and trained paraprofessionals with equids and their owners. In addition, there were anecdotal reports from the projects that the COVID-19 pandemic was negatively affecting equine welfare. Emergency feed relief interventions for equids in greatest need were organised in some countries by World Horse Welfare partners and, wherever possible, this was distributed within wider relief programmes organised by humanitarian organisations or local municipalities. Working equids are often invisible in government policy and funding, including during emergency events such as a hurricanes, volcanic eruption, and pandemics [12].

It became clear that NGOs needed to monitor the COVID-19 situation, to understand the effects of the pandemic on working equids and their owners, and to inform how to target interventions most effectively. The sustainability of interventions and the long-term economic effects on working equid-owning communities must also be considered.

The aim of this study was the assess the impact of the COVID-19 pandemic on the working equid community. The objective was to develop practical, robust, and reliable methodology to understand the effects of COVID-19 across equine welfare projects in 14 low- and middle-income countries during November and December 2020.

Through understanding how working equids and owners are affected by the COVID-19 pandemic, we can promote collaboration between NGOs and governments, reduce the detrimental welfare effects of future emergency events, and ensure that nobody is left behind.

## 2. Materials and Methods

A cross-sectional study was performed in 14 countries between 10 November and 31 December 2020, to investigate the effects of COVID-19 on working equid communities. The study was reviewed and approved by the Ethics Committee of the School of Veterinary Medicine and Science, University of Nottingham, England.

### 2.1. Survey Design

A pilot survey involving 24 open-ended questions was developed by World Horse Welfare. This survey was completed by 145 participants in Panama, Colombia, Nicaragua, Costa Rica, and Guatemala in May 2020. Results and feedback were reviewed. The survey was then developed by a University of Nottingham student (J.B. Jessica Burridge) into a more practical format that involved predominantly closed questions with pre-defined answers and optional free-text comments for some questions. Feedback was incorporated from World Horse Welfare International Programme Officers in the United Kingdom, World Horse Welfare Regional Coordinators, and World Horse Welfare partner-project staff. A revised version was piloted in the Colombia project, after which further minor adjustments were made. The final version included 38 questions that covered background information, equine-related income, equine welfare, general financial situation, aid/relief, and general impact of COVID-19. Some questions asked the participant to reflect on their situation at the time of the survey and their situation prior to the COVID-19 pandemic (e.g., January 2020). Apart from questions involving participant consent, country, and community, most questions were optional. The finished survey (Appendix A), which took approximately 20 min to complete, was available in Afrikaans, English, French, Khmer, Nepali, and Spanish. The survey schedule (Appendix A) is presented in two sections: Appendix A. Questions analysed in this study and Appendix A. Questions excluded from analysis; the numerical order is indicated and includes the total of 38 questions.

Survey responses were captured using an online platform SmartSurvey™ Ltd, Gloucestershire, UK [27], which allowed offline data entry that could be later uploaded once a data connection was available. The survey had the option of being carried out face-to-face, in remote, non-English-speaking communities, over telephone or dissemination on social media platforms. It was distributed to 14 countries worldwide.

### 2.2. Survey Logistics and Training

Safety of participants and project teams was a priority, and World Health Organization-recommended measures were taken to reduce the spread of COVID-19 including: carrying out surveys outside, physical distancing, wearing face masks, and disinfecting hands and contaminated materials [28]. Surveys were only carried out face-to-face if there were low infection rates in that area, if the project staff were asymptomatic, and if national/local restrictions permitted project activities at the time of carrying out the survey. Online video meetings were held with the Regional Coordinators to discuss logistical considerations including who would be conducting the survey, training required, should the survey be repeated, contacting the equid owners: face to face or over the phone, access to the internet, sample size, timeframe for data collection, equipment availability, and incentives. This led to individual projects filling out a form on logistical information.

The majority of projects indicated that they required training. As a result, training was delivered to all projects, regardless of prior survey experience. Training sessions, which were held via videocall, involved two UK-based researchers (A.G. and I.W.), the Regional Coordinators and project representatives who would be conducting the survey. Each representative received approximately 1 h of training relating to sample sizes, stratification by community, randomisation, how to pose the questions, how the platform worked (online/offline), and an explanation of all the survey questions. Emphasis was placed on achieving quality of results over quantity, as well as safety of all personnel, abiding by Government restrictions, and taking appropriate measures to reduce the spread of COVID-19. Further individual support on a case-by-case basis was subsequently provided by the UK-based researchers and project teams as required.

### 2.3. Participant Recruitment

Sample sizes for each project area were based on an estimate of the population of owners associated with each project and were calculated using Epitools [29], with an estimated true proportion of 0.3, and a confidence level of 0.95. The desired sample size for each project area was then divided among communities to achieve stratification by community (based on the population of owners in each community) within the project area. Randomisation was encouraged, with randomisation of sampling within the project population carried out by some projects, e.g., by surveying every 5th owner at the community visit (Lesotho, Guatemala, and South Africa). Most projects performed convenience sampling. Participants were recruited while the projects were conducting their usual community visits, or via telephone using an owner database. One project (Colombia) used the online link to disseminate the survey via social media. Overall, incentives were not provided, unless previously given and a precedent had been previously set for taking part in a survey. In Guatemala and Mexico, vitamins and minerals were provided as incentives and, in South Africa, 50 Rand supermarket and 50 Rand project coupons were provided (50 Rand is approximately USD 3.50).

The sole inclusion criterion was that participants must be equid-owners who had previously used, or were currently using, the project’s services. Participant information was verbally delivered and individuals were required to provide informed consent for the study. Data were anonymised, and the results were stored using password-protected software.

### 2.4. Country Context

Country context was obtained from each of the projects during the data collection period. This included current COVID-19 regulations and attitudes of people and additional factors such as season and adverse weather.

### 2.5. Data Analysis

The completed surveys were reviewed for duplicates, and any queries regarding the data followed up with the surveyors. It was found that there were some repeated phrases in the free-text comment boxes, however, the surveyors explained this was due to some participants giving the same answer for some questions. There were no duplicate surveys identified. Descriptive analysis was carried out on all survey results (frequency analysis and percentage). The closed question responses were automatically translated back to English by the survey platform. Comments in text boxes were translated by in-country project staff, and direct quotes from comments were included in the results to give context to some of the quantitative data, however, comments were not subject to in-depth analysis.

### 2.6. Feedback and Project Team Engagement

Weekly updates on the results obtained were sent to the project teams. Representatives from the projects who were conducting the survey were asked to fill out feedback forms and discussion was promoted via videocalls with two UK-based researchers (A.G. and I.W.) and the Regional Coordinators. The feedback included information about technical aspects of the survey, participant recruitment, interpretation of the survey questions, differences between administering the survey over the phone or face-to-face, whether there were any barriers to carrying out the surveys, and whether the teams needed further training.

## 3. Results

### 3.1. Demographics and Contextual Information (Q1, 2, 3, 5, 6, and 7)

A total of 1530 respondents consented to take part in the survey across 14 countries (Latin America, *n* = 847 (Colombia, *n* = 82; Costa Rica, *n* = 96; Guatemala, *n* = 211; Haiti, *n* = 4 Honduras, *n* = 136; Mexico, *n* = 139; Nicaragua, *n* = 102; Panama, *n* = 77); Africa, *n* = 332 (Lesotho, *n* = 74; Senegal, *n* = 147; South Africa, *n* = 43; Zimbabwe, *n* = 68); Asia, *n* = 351 (Cambodia, *n* = 242); Nepal, *n* = 109)). Seventy-eight per cent (1194/1523) answered the survey face-to-face, 21% (325/1523) by telephone, and <1% online/other (4/1523). There are a number of questions in the survey schedule (Appendix A) that have not been analysed in this study. This includes seven questions Q4, 10, 15, 19, 21, 35, and 38.

Restrictions (Figure 1) and case rates (Table 1) varied among the 14 countries involved in the study. “Country Context” data were obtained from all 14 countries (Appendix A). There were reports of considerable seasonal variation in Zimbabwe and Lesotho, with droughts and poor pasture. There was a hurricane that affected some countries in Central America in the lead up to the survey. In Nepal, brick kiln season was commencing, and Cambodia had seen some recent flooding. Perceptions on COVID-19 and associated in-country restrictions varied.

Mean household sizes and mean number of equids owned per respondent are displayed graphically, with regional differences seen and only small changes in number of equids owned before the pandemic and during the time of survey completion (Figure 2). The median number of equids owned varied from the mean, indicating that these data are not normally distributed: there was a median of one horse, zero donkeys, and zero mules owned per respondent, per region both prior to the pandemic and at the time of survey completion. Overall, each respondent owned a median of 2 equids both prior to the pandemic and at the time of survey completion. Median household sizes also varied from the mean: Latin America median five, Asia median five, and Africa median eight. Participant comments included: “During lockdown there was no work in the brick kiln, but nowadays the brick kiln [is] open and we added one new equine to earn more money” (Nepal); “I have sold my animals as I was desperate for cash” (Lesotho); “Five [equids] died due to drought” (Zimbabwe); and “Due to financial problems, we cannot afford sufficient feed for equines” (Nepal).

### 3.2. Type of Work Undertaken by Equids (Q8, 9, and 12)

The types of work undertaken by equids varied among regions (Figure 3) but there was little change in the types of work undertaken before the COVID-19 pandemic and at the time of survey completion. In Latin America, the two most common types of work were crop transport (before January 2020: 27% (468/1759) and November–December 2020: 27% (452/1670)) and freight transport (24% (414/1759) and 22% (374/1670)). In Africa, transport of people (26% (191/722) and 28% (191/687)) and water transport (17% (126/722) and 17% (114/687)) were most common, and in Asia, freight transport (37% (264/715) and 36% (266/736)) and crop transport (29% (206/715) and 30% (224/736)) were common. Globally, the most common uses were freight transport (25% (795/3196) and 24% (754/3093)), crop transport (23% (737/3196) and 24% (741/3093)), and transport of people (18% (572/3196) and 17% (530/3093)). Most comments included further clarification on types of work undertaken with their equid, but there were some comments related to why there have been changes: “Currently not working due to low number [three equids died] and poor condition” (Zimbabwe), “I use my donkey to collect firewood and sell it to community members but now after COVID-19 many people are no longer able to buy wood, hence the income generated per month has decreased” (Lesotho), “I normally plough for people… but few people afford to pay [and] money generated per month is lower” (Lesotho), and “We used to have village horse racing, but such activities are prohibited due to COVID-19 regulations, [so] money made from horse has decreased” (Lesotho).

Compared with pre-pandemic workloads, 50% (417/841) of equids in Latin America were working less and 40% (340/841) working the same amount at the time of survey completion. The most common reasons for change in workload were government enforcement/national restrictions (47%, 292/624) and change in demand (45%, 278/624). In Africa, the workload of 61% (201/330) of equids had reduced and this was mainly due to change in demand (85%, 234/274), government enforcement/national restrictions (61% 167/274), and local enforcement/regulations (60%, 165/274). In Asia, 73% (257/351) of equids were working less, largely due to a change in demand (66%, 230/350).

### 3.3. Financial Data (Q11, 13, 14, 24, 25, 26, 27, 28, 29, 30, 31, and 33)

Compared with pre-pandemic levels, monthly income derived from working equids and total household income had both decreased globally (76% (1130/1478) and 78% (1186/1519) respectively), with minor regional differences (Table 2). The results show that owner economy has not negatively impacted equid health (Table 2). A Senegal surveyor commented “I was happy to see that almost all of them [are making] the effort to keep the horses in good condition despite their difficult financial situation.”

At the time of survey completion, African owners most commonly reported that, if they were to sell their equid, its value would be the same (38% (124/330)) or less (37% (123/330)) compared to prior to the pandemic. The total cost of upkeep of their equid had generally stayed the same (63% (206/329)) or increased (31% (101/329)). In the Latin America, the most common response (42% 355/837) reported that if they sold their equid, it would be worth less than pre-pandemic. As in Africa, the total cost of equid upkeep in Latin America had generally stayed the same (54% (451/839)) or increased (36% (305/839)). In contrast, in Asia, the majority of owners said that equid prices were increasing (72% (252/350)) but, similar to the other regions, the total cost of equid upkeep generally remained the same (65% (229/351)) or increased (20% (71/351)).

Most equid owners across all projects saw a decrease in total income (Table 2), with their monthly outgoings and the cost of looking after their equine staying the same or increasing. As a result, many had to supplement their income in other ways. In Asia, this was typically via extra jobs (55% (190/347)) or using a money lender (25% (87/347)). Pre-pandemic, most Asian owners considered their financial situation to be the same as those in the same community without a working equid (71% (249/350)) but this decreased (66% (231/349)) at the time of survey completion. The percentage reporting that they were worse-off as a result of having an equid had risen from 5% (17/350) to 15% (54/349). In Latin America, 48% (391/811) of owners were not supplementing their income at the time of survey completion, while 24% (197/811) were doing so from government sources. Thirty-nine percent (324/835) of owners in Latin America considered themselves better-off as a result of owning a working equid both at the time of survey completion and before the COVID-19 pandemic (41% (339/826)). In Africa, at the time of survey completion, owners were supplementing their income via friends (50% (167/332)), extra jobs (46% (154/332)), and the money lender (42% (141/332)). Before the pandemic in Africa, 83% (274/331) considered themselves to be better-off as a result of owning a working equid, but this decreased to 37% (123/330) at the time of survey completion. At the time of survey completion, 45% (148/330) of owners in Africa considered themselves to be in the same financial situation as those in the same community without a working equid, with an increase in those considering themselves to be worse-off rising from 2% (6/331) to 14% (47/330).

The support schemes available to help owners during the COVID-19 pandemic varied among countries and regions. In Africa, 69% of respondents (230/332) had access to governmental assistance and 56% (187/332) to World Horse Welfare feed relief. Accessibility of schemes varied, with 21% of respondents (71/331) reporting that accessing them was very difficult, 19% (63/331) reporting that it was easy, and 19% (62/331) did not know about support schemes. Sixty-four percent (211/332) used these schemes, with 66% reporting them either beneficial or very beneficial (217/329). In Latin America, 31% (255/836) had access to World Horse Welfare feed relief, 24% (198/836) had access to “Other” relief, and 22% (183/836) had access to governmental assistance. Examples of “Other” relief include: “No schemes currently available” (Nicaragua) and “Equinos de Honduras (World Horse Welfare partner)” (Honduras). Ease of access to these schemes varied, with 31% (261/834) finding them easy to access, 20% (169/834) neither easy nor difficult, and 19% (160/834) did not know about support schemes At the time of survey completion, 58% (478/831) were using the schemes, and 67% (538/800) were finding them beneficial or very beneficial. Asia showed differing results, the most common answers were that 52% of respondents (183/351) were not aware of support schemes to help, 31% (108/351) had access to World Horse Welfare feed relief, and only 18% (64/351) having access to governmental assistance. Fifty-eight percent (178/307) did not know how easy these schemes were to access, 53% (185/351) had not used the schemes, and, likewise, 64% (221/348) did not know how beneficial the schemes were. In contrast to general support schemes, Asia had the greatest access to animal health support schemes (90% (316/351)), followed by Latin America (46% (386/836)) and Africa (17% (56/329)).

### 3.4. Equid Health (Q16, 17, 18, 20, 22, and 23)

Most owners reported that the health of their equid had not changed in association with the pandemic (68% (1034/1518)) (Table 2). However, the comments were quite variable: “They are almost not working, and they are fatter” (Costa Rica), “Underweight, not enough food” (Mexico), and “Unavailability of health care facilities during COVID-19 lock down [led to] many equine health problems” (Lesotho). When asked how happy/well their equid was at the time of survey completion, 70% (1060/1520) responded “Good”, with only 14% (207/1502) responding that their equid was “Poor” or “Very poor.” When asked about health problems that were present at the time of survey completion but not prior to the pandemic, Asian respondents reported the highest proportion of new health problems (23% (82/351)), followed by those from the Latin America (11% (88/837)) and Africa (4% (14/330)). Examples include general health problems: “Colic and lameness” (Nepal), “Hoof problems from lots of rain due to recent hurricanes” (Honduras), and “Corneal ulcer” (Mexico).

There were regional differences in changes in equid weight/body condition score (Table 2). Across Africa, droughts or limited grazing were reported by some of the projects and 62% (203/330) of owners saw decreased weight since the pandemic in their equids. While there were hurricanes in Central America in the lead up to the survey, 50% of respondents (417/833) reported that their equids had remained the same weight. Many owners in Cambodia reported having good grazing, with no comment on grazing condition in Nepal. There was a high prevalence of decreased workload seen in Asia (73% (257/351)) and increase in body condition score reported by 60% (212/351) of participants across Asia and specifically in Cambodia, 67% (166/242).

Across all countries, most participants reported the cost of equid-related services to have stayed the same (53% (760/1445)) or increased (32% (467/1445)) in association with the pandemic. Similarly, the availability of services largely remained the same (67% (970/1443)), although a small number of participants reported a decrease in availability of services (15% (219/1443)).

### 3.5. Other Considerations (Q34, 35, 36, and 37)

In Africa, 84% of owners (277/330); in Latin America, 38% (318/835); and in Asia, only 24% (83/351) reported that they would be comfortable having outsiders in their community. In Africa, concerning their own economy/livelihood, 53% (177/332) had moderate anxiety and 32% (105/332) severe anxiety. In Latin America, 44% (364/836) had mild anxiety and 37% (308/836) had moderate anxiety about their own economy/livelihood. In Asia, 44% (154/351) reported mild anxiety and 41% (144/351) severe anxiety about their own economy/livelihood.

When asked about positive aspects of the pandemic, there were a few comments: “Equids have more rest” (Panama), “Unity in the family” (Panama), and “Personal health awareness” (Mexico). However, most respondents replied that they had nothing positive to mention. Examples include: “A positive aspect? Nothing positive! I just want corona to leave [and] to let us work well!” (Nicaragua), “I have no idea about positive aspect, due to lockdown all our finances [had] decrease[d] and there is no income to provide good food for our family or horses” (Nepal), and “Misfortunes… [COVID-19 has] only brought death” (Nicaragua).

Feedback results from project staff are presented in Appendix A.

## 4. Discussion

Working equids play an essential role in the lives of the owners in the study population, through direct and indirect income generation and livelihood tasks including transportation of crops, goods, and people and providing tractive power in agriculture (Figure 3). This study demonstrates that the working equid community has been hit heavily by the COVID-19 pandemic: most equids are working less and equid-derived and overall household incomes are lower, while household expenses remain largely unchanged (Table 2). Currently, owner-reported negative health changes to their equids are variable and are low in frequency (Table 2). However, this may change with time. The situation needs close monitoring as the long-term economic consequences of the COVID-19 pandemic are likely to have serious impacts on low- and middle-income countries (LMICs) [4].

### 4.1. Demographics

In comparison to data from the Population Reference Bureau (PRB), household sizes are larger in this study [34]. For this comparison, the median is used as a summary statistic to compare with the PRB mean household sizes, due to the non-parametric data distribution in this study (Figure 2). The average household sizes for each region reported by PRB (based on the countries in this study) were 4.9 (Africa) compared to a median of 8 in this study, 4.4 (Asia) compared to median of 5 in this study, and 3.9 (Latin America) compared to median of 5 in this study [34]. It is widely recognised that larger household sizes and poverty are closely linked [35], and those relying on working equids are often the poorer citizens of LMICs [9,12].

Similar to FAOSTAT 2019 data, in this study, there is a larger proportion of donkeys owned in Africa vs. America and Asia [36]. However, donkeys and mules are under-represented in this study amongst all groups compared to FAOSTAT data [36]. Linking the number of equids owned with the household size, overall, this study found that a median of two equids support a median household size of five across the study population, therefore, a ratio of 2.5 people relying on 1 equid. There are regional differences, but as a whole, it shows that if an equid dies, is off work, or has decreased productivity due to an injury or sickness, the day-to-day lives/livelihoods of 2.5 people may be affected.

The numbers of equids owned did not change greatly between prior to the COVID-19 pandemic and the time of survey completion, which could be for a number of reasons: owners were not quick to sell their equids, equids were reproducing, owners were still relying on working equids for some tasks and household chores, owners were relying on equids for increased resilience as source of milk or meat, or perhaps owners were holding onto them due to emotional attachment despite decreased workloads. The reasons for this require further investigation.

### 4.2. Contribution of Equids to Owners’ Livelihoods

There were differences in the types of work undertaken by equids on a community, country, and regional basis. Such differences have not previously been documented in all of the countries involved in the survey. Working equids are involved in multiple tasks in the livelihoods of humans [9,12,20], and this study further emphasises their pivotal involvement. While the equids in this study are seeing decreased workloads, the type of work they are doing has not changed substantially in the short term. This could be due to the variable topography and equipment available, which may restrict the different tasks a working equid is able to do in individual settings in the short term. In Latin America, there were small decreases in equids being used for people transport, freight transport, crop transport, and racing (Figure 3). In Africa, the small number used for tourism and racing decreased, there was also a small decrease in percentage used for water collection, while there was an increase in ploughing (Figure 3). In Asia, there was a decrease in people transport and an increase in crop transport (Figure 3). Some of these small changes are not surprising with seasonal changes (Appendix A), the movement restrictions imposed due to the pandemic (Figure 1) [31], and the closure of markets, and restrictions on domestic and international travel (Appendix A).

### 4.3. Impact on Owner Income

It is clear from the results that the pandemic has had substantial effects on income within the working equid community (Table 2). Decreases in income for those living in LMICs attributed to the pandemic have been reported in other studies [37,38]. For example, Egger et al. (2021) reported that, across nine developing countries, a median of 68% respondents had experienced a reduction in income [37]. The World Bank has made grim predictions: “In the two decades since 1999, the number of people living in extreme poverty worldwide has fallen by more than one billion people. Part of this success in reducing poverty is set to be reversed due to the COVID-19 pandemic [4].” The current study found that owners from every region saw a decrease in income, with 76% reporting a decrease in equid-derived income and 78% reporting a decrease in total household income. These two statistics are closely linked, since many of the jobs that equids perform contribute directly or indirectly to owners’ overall income.

While income has decreased for many survey respondents, household outgoings have remained the same or increased for the majority, with the cost of keeping their equine also staying stable (Table 2). This is likely to lead to economic hardship, with many respondents reporting anxiety about their economy/livelihood. Many survey respondents have not seen the same level of income support that is available to those in high-income countries [39]. For example, in Africa, 69% reported that they had received governmental assistance, but this was lower in Latin America (22%) and in Asia (18%), where over half of Asians were not aware of any support schemes (52%). However, in terms of animal health support schemes, this was reversed, which was an interesting finding: in Asia, 90% said there were support schemes to help with animal health; Latin America, 46%; and Africa, 17%. Actions need to be taken to increase the support for owners in these areas, not only on an equine level but on a human level too, with the collaboration of humanitarian NGOs, animal welfare NGOs, and local and national governments.

### 4.4. Differences in Case Numbers, Restrictions, and Survey Results

Since the pandemic began, there have been substantial differences among countries in the numbers of cases and deaths associated with COVID-19 and in government restrictions (Figure 1 and Table 1). Reported case and death rates are not reliably comparable between LMICs as there are differences in how they are reported, and with limited testing available, positive cases may not be identified [40,41]. In this study, Lesotho is unique in that the first case was not identified until after movement restriction measures were implemented (Figure 1). Lesotho is enclaved within the border of South Africa and did not have COVID-19 testing capacity at the start of the pandemic, as such, these restrictions were a preventative measure, implemented after cases were identified in South Africa [42].

It could be hypothesised that those countries with relatively low case numbers and less severe restrictions might see less impact from the COVID-19 pandemic. Of the countries involved in our survey, Nicaragua has seen the fewest restrictions throughout 2020 (Figure 1) and low reported infection and death rates (Table 1), it is the poorest country in Latin America [43], and the country’s response may reflect this, due to lack of resources and the vulnerability of citizens. However, this response has been scrutinised and differs from approaches taken elsewhere [44,45]. The testing capacity of Nicaragua was also reported to be low [46]. Despite the lack of restrictions and low reported case numbers, the pandemic has still had a considerable impact on owners’ financial situation and livelihoods. Sixty-two percent equids are working less, largely due to change in demand reported by 67% of owners, 81% of owners have seen a decrease in equid-related income and 71% reported an increase in the cost of owning their equid(s). In this group, equid body condition was reported to have decreased for over half (54%). However, due to the season in Nicaragua, equids are usually seen in poorer condition at this time of year due lack of grass availability in the summer months, and seasonal hurricanes, so it is difficult to interpret this statistic (Appendix A).

### 4.5. The Inequality of the COVID-19 Pandemic

The crisis faced by LMICs as a result of COVID-19 differs from that faced by high-income countries [37,47]. For example, in lower-income countries, people have fewer household savings or governmental safety nets including compensation or income support to mitigate economic losses [39]. Furthermore, those on the lowest incomes in LMICs often work in the informal sector where job security is low [35,37,38]. These represent the jobs in which many of the working-equid community work. Additional social issues, such as the worsening of food insecurity, are a growing concern in LMICs [19,37,48,49,50]. Meanwhile, the direct impacts of SARS-CoV-2 on human health are considered to be lower in LMICs than high-income countries, due to younger populations [37]. In high-income countries, measures taken to reduce spread of disease, including physical distancing, are considered to have welfare value; these measures are considered to be less important and even detrimental in lower-income countries because of the economic trade-off [51].

Due to the pandemic, pre-existing global inequalities are worsening, with the rich:poor divide growing. Although populations are younger in LMICs than in high-income countries, healthcare systems are much poorer [52] and the healthcare essentials that are necessary to manage the COVID-19 pandemic have been in short supply [53]. Testing for COVID-19 has also been less available in LMICs and the total scale of disease in these areas cannot be calculated [40,41]. Finally, while vaccines provide a beacon of hope, procurement of vaccines in LMICs has been disrupted, with increased prices being charged to poorer countries and reduced accessibility [54,55,56].

### 4.6. Impact on Equine Welfare

Animal welfare is facing considerable collateral damage from the pandemic [57]. The World Organisation for Animal Health (OIE) predicts the economic recession may affect owners’ ability to afford feed and healthcare for their animals [58]. This is a worry echoed in this study. COVID-19 has already been shown to have a negative effect on equine welfare in high-income countries, with impacts on horse abandonment, equine business, equid management practices, and accessibility of veterinary/paraprofessional services [58,59]. Welfare issues specifically affecting production animals include a slowdown in slaughter and consequent overcrowding, longer transport times to slaughter, on-farm depopulation, animal feed shortages, and decreased supply of medication and veterinary services [58,60,61,62]. There is a current paucity of published data on working equids in LMICs for direct comparison of results.

In this study, it was considered that regional variations in owner-reported equine body condition was multifactorial and affected by the season and adverse weather. In Africa, 62% of owners reported a decrease in equid weight compared with prior to the pandemic, impacted by the dry season and droughts. In the short term, it appears that with 57% of equids working less, there is greater opportunity for weight gain and recovery from health problems, with most owners reporting that their equids’ health had stayed the same (68%) and that their equid was “good” when questioned how happy/well their equid was (70%). In contrast to other studies, reduced accessibility of equine services in association with the pandemic was not a major finding in this study, with 67% reporting service availability was unchanged. However, it was found that costs of equine services were increasing in some cases (32%), with approximately half reporting the costs remained the same, which may have longer-term impacts on equine welfare if owners are unable to afford increasing prices alongside decreasing income.

### 4.7. Strengths, Limitations, and Future Research

The survey development process involved multiple iterations and two pilots, with input from a broad range of perspectives. Although the aim was to develop a survey that minimised bias, it is likely that bias remained. For example, although each interviewer received the same training, individuals had differing levels of previous experience in carrying out surveys and interviewer bias is therefore likely. In addition, although attempts were taken to randomise the sampling within the project population, this occurred in some but not all participating projects and the resultant convenience sampling is likely to be associated with selection bias. Third, although a priori sample size calculations were undertaken, not all projects were able to obtain a sample representative of their project population within the data collection period due to the COVID-19 pandemic, other adverse events, and time constraints. However, it is contended that the total number of respondents (1530 across 14 countries within three global regions) provides a useful insight into the current situation for working equids and their owners across the World Horse Welfare International Programme. The programme has more projects in Latin America versus Africa and Asia, hence the larger population of this group. Direct comparison of data from different regions is challenging due to the differences in sample size and convenience sampling, but it is anticipated that statistical testing of data from future surveys will be of merit for longitudinal comparison of results within countries and regions. It should also be noted that while the results are described regionally, results are not representative of the entire region, rather, the World Horse Welfare projects in that region. Finally, it is acknowledged that there will be seasonal bias, as illustrated in the country context (Appendix A), and that the timelines and effects of COVID-19 infection rates and restrictions varied by country (Figure 1, Table 1, and Appendix A). These factors will inevitably influence the results.

Some limitations in this study will inform considerations for different approaches in future surveys. Most questions were well-understood, but it was clear from the responses and feedback that some questions require modification, with better framing and improved translation before use in future surveys, so that all results can be reliably analysed. This meant some questions featured in the survey were excluded from evaluation in this study. In future surveys, emphasis will be placed on ensuring that surveyors fully understand all questions, so that they can explain any challenging questions to participants. Based on feedback to keep the survey short, certain demographics such as gender and age of the respondent were not included. These would be useful in further surveys for comparison with other studies and to determine the involvement of women involved in the projects, which can be under-represented as equid-owners, but often contribute significantly to the care of equids [63,64]. This was a survey of owners’ opinions, and in future surveys, including an objective welfare assessment undertaken simultaneously by trained professionals, would provide insightful triangulation of results and ensure that welfare issues are not missed. Those no longer owning equids were less likely to attend the equid-based community activities, at which many of the surveys were carried out, therefore, this population may have been missed. Equine owners attending the project activities are usually more motivated to improve equine welfare than those choosing not to attend activities, so equine welfare indicators in this study may be better than the general population of equids in the same area. These two missed populations could be sought out in future surveys for comparison of results.

Feedback obtained from project staff will be used to adapt the survey for future use (Appendix A). There was variation in how easy it was for the projects to carry out the survey and sharing the overall feedback between projects promoted learning opportunities. While the risks of COVID-19 need to be taken into consideration, it was clear that face-to-face surveys were easier. Using the face-to-face methods over telephone or web surveys allows access to isolated communities who may not be captured in other studies [37,65]. There were useful comments on challenging questions specific to some counties, such as in Senegal, where owners were less comfortable talking about household size, while income-related questions were deemed as a challenging question for multiple projects. All feedback will be reviewed, with specific questions reviewed and further training provided prior to future surveys.

This survey was designed with the intention of informing Word Horse Welfare and partner organisations about the effects of, and the response to, COVID-19 within the working equid community. The survey results are concerning and indicate that action must be taken. Each project is currently reviewing results on a community and country basis, developing plans, and targeting the key areas of concern highlighted in the study. Follow-up surveys will be used to assess whether the interventions made are improving the situation. A monitoring and evaluating approach will be taken now that baseline data has been gathered in this study [6]. It is clear from the results that the long-term impacts of the pandemic need to be assessed in the working equid community. These methods have been valuable to collect key information during the COVID-19 pandemic and with adaptations, will be appropriate for use in future emergency events.

### 4.8. Conclusions

This study explored the short-term effects of the COVID-19 pandemic on the working equid community across World Horse Welfare’s International Programme. The findings showed that there were marked effects, with equid owners reporting decreased equid workload, decreased equid-derived income, and decreased overall household income, while household expenses remained largely unchanged. Equid welfare has been impacted variably. The situation requires close monitoring, including follow-up studies exploring the long-term impact of the pandemic.

The evolution of the COVID-19 pandemic fundamentally demonstrates the importance of One Health approaches and the pandemic itself has One Welfare consequences. Collaboration among animal welfare NGOs, governments, and humanitarian organisations is needed to overcome deep-rooted issues, exacerbated by the pandemic, involving the marginalised working equid community. There is an opportunity to learn from this situation, with the promotion of long-lasting change. We must build back better and ensure that systems to protect the health and welfare of equids and their owners are resilient.

## Figures and Tables

**Figure 1 animals-11-01363-f001:**
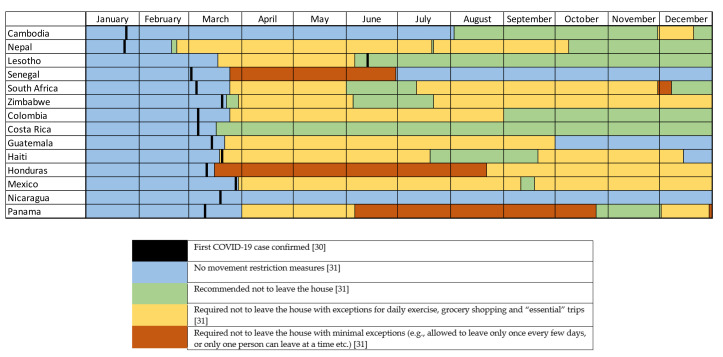
First confirmed COVID-19 case and associated movement restrictions across the 14 countries involved in the survey, incorporating data from Worldometer (first confirmed case) and Our World in Data (movement restrictions) [30,31].

**Figure 2 animals-11-01363-f002:**
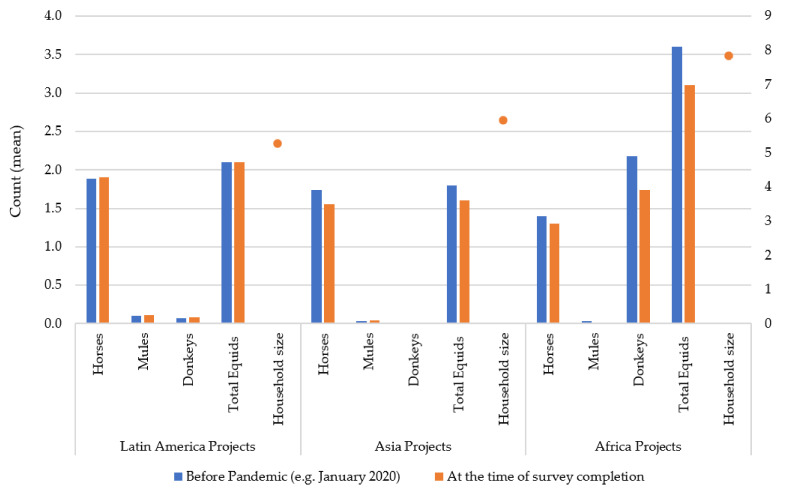
Regional differences for mean number of equids owned per respondent, before the pandemic and at the time of survey completion, measured using the primary axis. Regional differences for mean number of people living in a household per respondent, at the time of survey completion, represented by the orange dots and measured using the secondary axis.

**Figure 3 animals-11-01363-f003:**
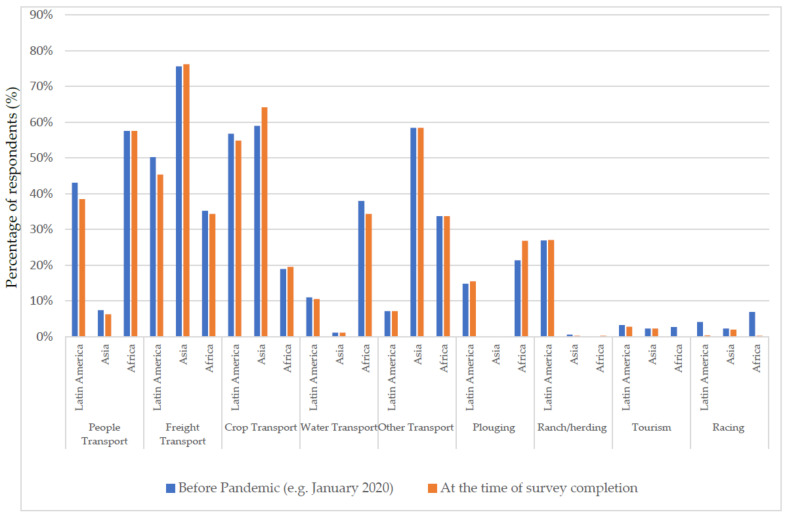
Types of work carried out by working equid, as reported by owners.

**Table 1 animals-11-01363-t001:** Total numbers of cases and deaths recorded due to COVID-19 on 31 December 2020 from Our World in Data [32,33], ordered on descending order of number of cases.

Country	Total Number of Cases	Total Number of Deaths
Colombia	1,640,000	43,213
Mexico	1,430,000	125,807
South Africa	1,060,000	28,469
Nepal	260,593	1856
Panama	246,790	4022
Costa Rica	169,321	2185
Guatemala	138,012	4813
Honduras	121,827	3130
Senegal	19,140	410
Zimbabwe	13,867	363
Haiti	9999	236
Nicaragua	6046	170
Lesotho	3094	51
Cambodia	378	No data

**Table 2 animals-11-01363-t002:** Modal owner-reported changes in income, outgoings, and equid health and workload, by region at the time of survey completion versus pre-pandemic. Modal response in bold.

Variable	Response	Latin America Projects	Africa Projects	Asia Projects
**Change in equid workload**	Decreased	**50% (417/841)**	**61% (201/330)**	**73% (257/351)**
	No change	40% (340/841)	23% (77/330)	24% (83/351)
	Increased	8% (70/841)	11% (35/330)	1% (2/351)
	Not working	2% (14/841)	5% (17/330)	3% (9/351)
**Change in monthly income from equid**	Decreased	**73% (601/824)**	**79% (238/303)**	**83% (291/351)**
	No change	24% (195/824)	19% (58/303)	15% (54/351)
	Increased	3% (28/824)	2% (7/303)	2% (6/351)
**Change in total monthly household income**	Decreased	**73% (601/824)**	**79% (238/303)**	**83% (291/351)**
	No change	24% (195/824)	19% (58/303)	15% (54/351)
	Increased	3% (28/824)	2% (7/303)	2% (6/351)
**Change in cost of upkeep of equid**	Decreased	8% (63/839)	5% (17/329)	14% (50/351)
	No change	**54% (451/839)**	**63% (206/329)**	**65% (229/351)**
	Increased	36% (305/839)	31% (101/329)	20% (71/351)
	Did not know	2% (20/839)	2% (5/329)	0% (1/351)
**Change in monthly household expenses/outgoings**	Decreased	25% (205/834)	20% (66/327)	23% (79/351)
	No change	**38% (316/834)**	**52% (171/327)**	**46% (160/351)**
	Increased	38% (313/834)	28% (90/327)	32% (112/351)
**Change in equid health**	Deteriorated	12% (104/837)	8% (26/332)	2% (8/349)
	No change	**61% (513/837)**	**82% (273/332)**	**71% (248/349)**
	Improved	11% (92/837)	6% (21/332)	26% (89/349)
	Did not know	8% (65/837)	2% (8/332)	1% (2/349)
	Other	8% (63/837)	1% (4/332)	1% (2/349)
**Change in equid weight/body condition score**	Decreased	24% (197/833)	**62% (203/330)**	5% (17/351)
	No change	**50% (417/833)**	27% (88/330)	35% (122/351)
	Increased	23% (192/833)	10% (32/330)	**60% (212/351)**
	Did not know	3% (27/833)	2% (7/330)	0% (0/351)

## Data Availability

Link to publicly archived dataset generated and analysed during the study: DOI: 10.17639/nott.7108.

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
