# Peer review of "The Impact of COVID-19 on the Working Equid Community: Responses from 1530 Individuals Accessing NGO Support in 14 Low- and Middle-Income Countries"

_animals, 2021, doi:10.3390/ani11051363_

Round 1

Reviewer 1 Report

Manuscript ID animals-1187116

The impact of COVID-19 on the working equid community: responses from 1,530 individuals accessing NGO support in 14 low- and middle-income countries

The authors described an extremely interesting problem regarding horse welfare during the COVID-19 pandemic in 14 low- and middle-income countries. Noteworthy is the extensive consideration of the problem in terms of the development of the pandemic and the introduction of various restrictions on the territory of individual countries. The problem has been widely discussed in terms of economic and social factors. It is very important that the authors point to the problems related to the generally difficult situation of people dealing with horses, which was aggravated by the pandemic. I believe that this is a very interesting topic that should be continued.

Author Response

Dear Reviewer,

Thank you for reading and commenting on the manuscript. We appreciate the positive comments and your insights on this work. This is a topic we will continue to explore and learn from, and endeavour to understand the solutions that may be developed through collaboration with other organisations. 

Yours sincerely,

Dr Isabella Wild MRCVS

Reviewer 2 Report

The topic of this paper is interesting, useful and relevant for this journal Frontiers Veterinary Science. It is well written and investigated the effects of COVID-19 on working equid communities with a multi-language survey.  The methodology developed is interesting.  I recommend no revisions.

Author Response

(The authors gave the same response as above.)

Reviewer 3 Report

A much needed study which provides a valuable benchmark to showcase the initial impact of the Covid-19 pandemic on working equids and their owners, which will have enduring value as the ongoing impact of the pandemic is assessed in these vulnerable populations.

Simple summary: concise and good summary provided

Line 12: suggest replacing instituted with implemented

Abstract: again concise and good summary provided

Introduction:

Good introduction to study provided which sets scene and need for the research to underpin human and equid welfare well.

Line 69: remove is to improve sentence flow

Line 72: add comma after (NGO)

Materials and methods:

Comprehensive and detailed overview of methods provided.

Line 179: please outline the methods used to analyse / translate comments

Results:

A lot of information here across multiple regions, which is presented well.

Figure 3: add notation to identify what orange dots represent on figure 3 to figure legend

Discussion:

Key results well evaluated and used to underpin areas where action is needed to safeguard equid and human welfare; complemented by good consideration of limitations and strengths within study design presented.

Line 409: would be worthwhile to consider the impact of the pandemic on differences across the countries / regions assessed in a little more depth here especially if some of these data represent a ground line for the use of equids in specific areas

Conclusions:

Currently I feel the conclusion presented would be better suited as a final paragraph within the discussion and the authors could then summarise their key results and take home messages to the reader here, concluding with a call to arms for future work in this field.

Author Response

Dear Reviewer,

Regarding: The impact of COVID-19 on the working equid community: responses from 1,530 individuals accessing NGO support in 14 low- and middle-income countries

Thank you for reading the manuscript and providing insightful comments on how the manuscript could be improved. A table of corrections is provided (please see attachment) to detail the changes that have been made following your suggestions. Thank you again for your time and help with this. 

Yours sincerely,

Dr Isabella Wild MRCVS

Reviewer 4 Report

Dear authors, this is a very nice study unravelling some of the constraints that owners of working equids have been faced to during the pandemic. I have some suggestions that I hope can improve the manuscript.

In relation to keywords I suggest not ussing not using acronyms such as LMIC, and I would delete charity and sustainable development goals, I think with sustainable development is enough.

Objectives, the authors present 2 objectives, but I think the first one is not covered in sufficient detail in terms of methodology in this manuscript, maybe having both aims is a little ambitious. I would keep one objective aiming to have a first approach to understanding the effects of the COVID-19 pandemic accross working equids associated to the projects.

The material and methods section requires more detail to fully comprehend it. For example I understand the full questionnaire has 38 questions, but not all are presented in this manuscript, this is a little confusing and maybe only the questions analysed should be included.

Figure 2 I would suggest directly stating in the leyend that the Data is according to Worldometer and Our World in Data. I would also add the year and at least for me it is difficult to understand why in the case of Nicaragua there is no first covid case confirmed, although in Table 1 the number of cases is provided, or why in some countries the first confirmed case is after movement restrictions, this should be later on discussed.

L216 I believe it should be affected instead of effected.

Figure 3. I suggest moving the household size to a secondary axis, that would allow making more visible the mean number of horses, donkeys and mules per house. If in some cases a household has horses and donkeys, or mules maybe it would be appropriate to add the mean number of equids by household.

L242 replace : with ;

Table 2 was a little difficult to read for me because in some cases the percentage of decrease is given, but in others the no change or the increase. If this are the results of a close ended question I suggest including the results of all options. In the same line, in the case of change in equid health, and change in equid BCS, the change is according to the owner's perception? or was this assessed by the project professionals?

Line 279. it says most owners in Africa, but the value is 38% and 37%, also please check that it says 124/33 and 123/33, that 33 does not seem to be right.

Line 405-407. can this also be associated to the price decrease or because equids still help with shores within the household? or can still increase resilience as a source of milk, meat or offsprings?

Line 460-462. Nicaragua is probably the poorest country in Latin America, can this lack of restrictions be associated to the vulnerability of citizens and the impossibility of the government to provied help in case of a lockdown?

L465-467. By this group do you mean Nicaragua? or Latin America? maybe the characteristics of the season should be defined to better understand this part.

L552. I suggest including something in relation to the fact that owners that attend the clinics are usually more responsible and this could signify that the results are better than what we could find for owners not involved in welfare programmes.

References: Please check the instructions for the reference list.

Author Response

Dear Reviewer,

Regarding: The impact of COVID-19 on the working equid community: responses from 1,530 individuals accessing NGO support in 14 low- and middle-income countries

Thank you for reading the manuscript and providing insightful comments on how the manuscript could be improved. A table of corrections is provided (please see attachment) to detail the changes that have been made following your suggestions. Particular attention has been paid to the results section, in light of the comments. Thank you again for your time and help with this. 

Yours sincerely,

Dr Isabella Wild MRCVS

Round 2

Reviewer 4 Report

I would like to thank the authors for addressing all my suggestions. Hopefully they also feel that the manuscript is clearer.

I have some minor suggestions:

L236, it would be better to use physical distancing rather than social

L315 I suggest adding the aprox value of 50 Rand into US dollars so the reader can get an idea. 50 Ran (aprox 3.5 US dollars??)

L327 when the authors mention duplicates, it is not clear for me if you mean that the same owner was interviewed twice?

Table 2. The heading of the first column "time of survey completion vs pre-pandemic" does not seem right. I suggest adding a second column with the options, that way you can leave only numbers under the region columns. It is not necessary to mention all countries in each region in the column headings. I think these changes will make the table more readable.

Suggested Table example:

Table 2. Modal owner-reported changes in income, outgoings, and equid health and workload, by region at the time of survey completion versus pre-pandemic. Modal response 523 in bold.

Variable

Response

Latin America projects

Africa projects

Asia projects

Change in equids’ workload

Decreased

50% (417/841)

31% (201/330)

73% (257/351)

No change

40% (340/841)

23% (77/330)

24% (83/351)

Increased

8% (70/841)

11% (35/330)

3% (9/351)

Not working

2% (14/841)

5% (17/330)

1% (2/351)

Also in the example please check the 3% and 1% because in the table in the manuscript increased and not working are inverted in the last column (Asia)

In the references section add all authors to references 3, 4, 8, 9, 13, 16, 50, 52, 53, 55, 58, 61.

Reference 47 is missing in the reference list

Author Response

Dear Reviewer, 

Thank you for these constructive comments, they are useful and make the manuscript clearer. There are differences in the line numbers with the downloaded latest manuscript, and I have updated the line numbers in the table of corrections to make this clearer. Please find table of corrections attached. 

Yours sincerely,

Isabella Wild
